# Gut Microbiota between Environment and Genetic Background in Familial Mediterranean Fever (FMF)

**DOI:** 10.3390/genes11091041

**Published:** 2020-09-03

**Authors:** Agostino Di Ciaula, Alessandro Stella, Leonilde Bonfrate, David Q. H. Wang, Piero Portincasa

**Affiliations:** 1Clinica Medica “Augusto Murri”, Department of Biomedical Sciences and Human Oncology, University of Bari “Aldo Moro”, 70124 Bari BA, Italy; agostinodiciaula@tiscali.it (A.D.C.); leonildebnf@gmail.com (L.B.); 2Section of Medical Genetics, Department of Biomedical Sciences and Human Oncology, University of Bari “Aldo Moro”, 70124 Bari BA, Italy; alessandro.stella@uniba.it; 3Department of Medicine and Genetics, Division of Gastroenterology and Liver Diseases, Marion Bessin Liver Research Center, Einstein-Mount Sinai Diabetes Research Center, Albert Einstein College of Medicine, Bronx, NY 10461, USA; david.wang@einsteinmed.org

**Keywords:** amyloidosis, colchicine, inflammasome, interleukin-1b, *MEFV*

## Abstract

The gastrointestinal tract hosts the natural reservoir of microbiota since birth. The microbiota includes various bacteria that establish a progressively mutual relationship with the host. Of note, the composition of gut microbiota is rather individual-specific and, normally, depends on both the host genotype and environmental factors. The study of the bacterial profile in the gut demonstrates that dominant and minor phyla are present in the gastrointestinal tract with bacterial density gradually increasing in oro-aboral direction. The cross-talk between bacteria and host within the gut strongly contributes to the host metabolism, to structural and protective functions. Dysbiosis can develop following aging, diseases, inflammatory status, and antibiotic therapy. Growing evidences show a possible link between the microbiota and Familial Mediterranean Fever (FMF), through a shift of the relative abundance in microbial species. To which extent such perturbations of the microbiota are relevant in driving the phenotypic manifestations of FMF with respect to genetic background, remains to be further investigated.

## 1. Introduction

The gut develops as a natural ecosystem hosting a complex polymicrobial community, referred to as microbiota. The microbiota can undergo major changes during healthy status and diseases [1,2]. The resident symbiotic microorganisms have progressively adapted to several factors, i.e., local environment, the host immune responses, antibiotic therapies, and several other conditions [3,4,5]. In the human gut, there are thousands of different microbial species [6], possibly conditioning health and disease. Examples include inflammatory bowel disease [7], irritable bowel syndrome [8], periodontal disease [9], atherosclerosis [10], rheumatoid arthritis [11], diabetes mellitus [12], obesity [13], allergy [14], and colonic cancer [15].

The exact role of intestinal microbiota in other conditions characterized by recurrent genetically-driven auto-inflammatory diseases is still to be comprehensively determined. Familial Mediterranean Fever (FMF) is an example of monogenic autoinflammatory disease due to *MEFV* gene pathogenic variants that lead to a dysfunctional hyperactive state of the pyrin protein eliciting proinflammatory cytokine release and pyroptosis (cell death).

Studies focusing on human twins [16], murine quantitative trait loci [17], and genome-wide associations [18], suggest that the host genomic profile can shape the composition of gut microbiota. This step becomes mainly host genotype-dependent [19,20]. The knowledge of mechanisms linking the genome with the abundance and the composition of gut microbiota, however, deserves further study. A main interfering factor is the role played by gene-environment interactions [21]. This topic is of great interest, since gut microbiota are essential for human health and in maintaining a systemic homeostasis. In fact, several pathological conditions have been linked to variations in the amount and/or composition of the microbiota. Instead, the therapeutic manipulation of gut microbiota is a possibly relevant and innovative tool, in particular during diseases characterized by a chronic inflammatory status. Based on genetic (i.e., monogenic disease) and phenotypic characteristics, FMF patients can represent an interesting model to assess the role of genome and gene-environment interactions, in modifying the composition of gut microbiota. A comprehensive view of these dynamics could be useful in the management and prevention of acute attacks and complications in FMF patients.

Thus, it appears of great relevance to comprehensively review the major features of gut microbiota in the human gastrointestinal tract, highlighting the emerging relationship between variations in the relative abundance of microbial species and FMF.

## 2. Familial Mediterranean Fever

FMF is a rare monogenic autoinflammatory disease which belongs to the group of “periodic fever syndromes.” The estimated number of subjects with FMF in the world is 150,000, making FMF the most common autoinflammatory disease worldwide [22]. FMF occurs more often in individuals of Jewish, Armenian, Turkish, North African, and Arab descent, and living in the Mediterranean basin. Cases of FMF also occur in different populations such as Greeks, Italian, and even Japanese [23,24]. Many patients with FMF describe their first attack in early childhood, i.e., before the ages of 10 (65%) and 20 years (90%). Depending on genetic penetrance and phenotypic characteristics, however, the initial attack can occur in subjects aged older than 50 years [23] with extremely delayed onset represented by one case diagnosed at the age of 86 [25].

FMF is mainly diagnosed for its typical clinical presentation, combined with ethnic origin, family history, and genetic assessment [22,26,27,28,29]. Genetic testing can help but it is challenging, since 377 different *MEFV* variants have been reported thus far (Infevers: an online database for autoinflammatory mutations. Available at https://infevers.umai-montpellier.fr/ accessed 26.08.2020) [30,31,32,33]. Whereas some *MEFV* variants appear as clearly pathogenic, many variants are common in the general population and some others have still an unknown significance in causing the disease [34]. In line with such difficulties in classification of *MEFV* variants, recent efforts have tried to develop novel classification tools based on machine learning, which led to improvement in *MEFV* variants classification [30,35,36]. On chromosome 16 (16p13.3), the *MEFV* gene (made of 10 exons) encodes for a 781-amino-acid ~95kDa protein named pyrin (also referred to as “marenostrin,” TRIM20), a pattern recognition receptor (PRRs) [37,38,39,40,41]. Pyrin is part of the complex molecular platforms involved in the response of the innate immune system and related cells, originally designed as first-line, fast response to components of pathogenic bacteria. Cells involved in the innate immune system response are monocytes, macrophages, dendritic cells, and neutrophils (myeloid lineage), which express a variety of PRRs. PRRs, in turn, detect pathogen-associated molecular patterns (PAMPs). The family of Toll-like receptors (TLRs) are membrane-bound PRRs sensing PAMPs in the extracellular milieu and in different types of intracellular endosomes [42]. TLR activation is associated to the expression of proinflammatory factors, which induce cytokine release, i.e., NF-κB. Cytosolic pathogen recognition sensors are the family of nucleotide-binding domain leucine-rich repeat (NLR) proteins, namely NLRP1, NLRP3, NLRP7, and NLRC4, the protein absent in melanoma 2 (AIM2), and pyrin [43]. These cytosolic sensors detect pathogens and endogenous danger-associated molecular patterns (DAMPs) which trigger the intracellular formation of multiprotein complexes, i.e., inflammasomes [44].

A common feature of inflammasomes is their capability to mediate the activation of caspase-1 and, subsequently, to promote the release of the proinflammatory cytokines IL-1β and IL-18. Another step is represented by the activation of inflammatory cell death via pyroptosis, with cellular swelling and lysis, at variance with apoptosis. Pyroptosis requires the caspase-1-mediated cleavage of gasdermin D (GSDMD), translocation of the fragment *N*-terminal pore-forming domain to the cellular membrane, and release of pro-inflammatory cytokines [45,46]. Notably, the direct binding of lipopolysaccharide (LPS) to caspase-4 and 5 in human cells (caspase-11 in mouse cells) will also result in caspase oligomerization, cleavage of GSDMD, and pyroptosis. In general, pyroptosis appears to amplify the defending immune responses following infections [45].

In a normal situation, pyrin senses the inactivation of the small RhoA guanosine triphosphatase (RhoA GTPase), triggered by bacterial toxin, and this step leads to the activation of numerous signal transduction pathways resulting in the formation of a multi-protein complex (inflammasome). Pyrin binds to several effector proteins, such as the serine/threonine-protein kinases PKN1 and PKN2 and actin-binding proteins. RhoA activation is associated to PKN-mediated phosphorylation-dependent pyrin inhibition. The inflammasome also contains the bridging molecule ASC (apoptosis-associated speck-like protein containing a caspase recruitment domain) and the protease caspase-1 [38,44,47,48] (Figure 1).

The activated inflammasome will then govern the steps of pyroptosis, i.e., a pro-inflammatory cell death mode which relies on innate immune response by myeloid cell lineage with release of pro-inflammatory cytokines IL-β1 and IL-18 [45,47,59]. Mutations in the *MEFV* gene are associated with impaired function of pyrin, which becomes insensitive to the microtubule control of this process [41]. Consequently, raised serum levels of IL-1β and increased inflammation will occur (Figure 2A,B).

On the clinical ground, FMF consists of periodic recurrent febrile attacks, serositis at multiple body sites manifesting with pain (abdomen, joints, chest), erysipelas-like dermatitis (meaning limited erythematous skin rash), myalgia, arthralgia, and acute pericarditis. Depending on *MEFV* variants involved, symptoms may appear during the pediatric age, with episodes of pain and fever usually resolving within 2–3 days (Figure 3).

Acute attacks of FMF produce elevation of serum indices of inflammation, including a raised count of white blood cells (especially neutrophils), increased levels of C-reactive protein (CRP), erythrocyte sedimentation rate (ESR), fibrinogen, and serum amyloid A (SAA) protein. Depending on *MEFV* variants and intensity of attacks, long-term complications may include secondary (AA) amyloidosis leading to asymptomatic proteinuria, nephrotic syndrome and end-stage kidney disease, small bowel obstruction due to recurrent attacks of peritonitis and adhesions, and even infertility, especially in female patients due to fallopian tube obstruction [24,34,60]. The M694V variant, located in exon 10 of the *MEFV* gene, causes the most severe disease, in patients either homozygous or compound heterozygous for M694V [61,62]. The same is true for M694I and M680I variants, while R761H (or E148Q/R761H) has lesser penetrance and causes milder symptoms [24,35]. The FMF therapy focuses on preventing acute attacks, and minimizing subclinical inflammation in between attacks. In the most clinically-evident cases, the appropriate therapy will also prevent development and progression of amyloidosis. All guidelines suggest colchicine as the initial treatment of FMF, to be started as soon as a diagnosis of FMF is established and continued indefinitely. In addition, colchicine is effective as a prophylactic treatment for FMF attacks, and for this purpose, all patients should start with colchicine, regardless of the frequency and intensity of attacks [24,28,63,64] (Figure 2A and Figure 4). A subgroup of patients defined as resistant or intolerant to colchicine needs treatment with alternative biological agents targeting IL-1 inhibition. The human immunoglobulin (IgG) antibody canakinumab targets IL-1β and is effective in FMF [49,51,65,66,67,68] (Figure 4 and Figure 5).

## 3. Gut Microbiota

Gut microbiota consists in a huge collection of microbes in the human gut. The human microbiota varies according to birth mode, age [69], diet and lifestyle, and is essential in preserving the integrity of the mucosal barrier [70]. Furthermore, gut microbiota protects the intestinal epithelial cells and contributes to the immune system [71] producing antimicrobial peptides and immunoglobulins [72,73,74].

### 3.1. Development of the Gastrointestinal Microbiota

The interplay between microbiota and the gut begins very early in human life.

The in utero environment is not sterile, with a maternal-fetal transmission of microbiota occurring early, during pregnancy [75,76].

The genome of the infant may affect gut microbial colonization but, on the other hand, the composition of gut microbiota depends on a number of maternal factors acting before birth (dietary habits, gestational age, smoking, obesity, antibiotic therapy during pregnancy) [5,77]. At birth, the newborn acquires further bacteria from the mother and the external environment [78,79]. Facultative aerobes anticipate the increase in strict anaerobes [80]. *Bifidobacteria* are more prevalent during breast- or formula feeding [81,82,83]. Feeding with solid food in the infant is followed by changes in the composition of gut microbiota, with features resembling those of the adult [84]. According to this scenario, the gut microbiota stabilizes after the first 12 months of life.

### 3.2. Characterization of the Gut Microbiota

Most of the gut bacterial species cannot be cultivated [81,85,86,87,88], while culture-independent molecular approaches provide information on composition and diversity of the gut microbiota [89,90,91,92,93]. Techniques include sequencing and phylogenetic micro-arrays [94,95], metagenomic assembly, annotation and comparison [3], genome-resolved metagenomics. Metaproteomics, metabolomics and metatranscriptomics require methodologies that are more sophisticated. The Human Microbiome Project (HMP) employed the 16S and metagenomic profiling to investigate the microbial communities from multiple body sites of healthy individuals, the relationships between disease occurrence and microbiota, and to identify a standardized dataset [96,97]. The human intestinal microbiota contains about 3.3 million microbial genes, and this amount is about 150 times greater than the human genome [98].

### 3.3. Main Features of Bacterial Communities in the Human Intestine

During health, the dominant phyla are Bacteroides and Firmicutes [99,100] with a minor presence of Spirochaetes, Proteobacteria, Verrucomicrobia, Actinobacteria, Lentisphaerae and Fusobacteria [19,94]. Microbial density increases in oro-aboral direction, raising from 10^4^–10^8^ cells in the jejunum-ileum to 10^10^–10^12^ cells at the level of the colon and in feces [3,101]. Different microbiota ecosystem changes depend on pH in the intestinal lumen, redox potential, type and amount of nutrients, gastrointestinal motility and secretions [3] (Figure 6).

In the jejunum predominate few species of bacteria (i.e., *Enterococci, Lactobacilli*, oral *Streptococci* and other gram-positive aerobic or rare facultative anaerobes from the oropharynx), with concentrations reaching 10^4^ CFU/mL of jejunal content. The microbiota at this level is sensitive to and controlled by physiological factors, including the presence and type of bile acids, the intraluminal pH, pancreatic secretion and gastrointestinal motility [101]. The ileum becomes a more favorable environment for microbial growth where *enterobacteria* and other coliforms reach a concentration of 10^9^ CFU/mL [101]. Again, peristalsis in the small intestine and appropriate gastric acid secretion prevent qualitative, quantitative and/or topographic changes of the intestinal microbiota, namely the bacterial overgrowth [107]. The colon hosts up to 10^12^ CFU/mL of numerous bacterial species, primarily anaerobes (*Bacteroides, Clostridium, Bifidobacteria*, *Lactobacilli*) [108]. In the feces, anaerobic bacteria predominate with a great diversity (range 3000 to 5000 species) [109]. About 90% of the fecal mass is made of bacteria and optimal temperature and viscosity can facilitate bacterial growth. The human gut hosts three bacterial enterotypes, not country- or continent-specific. These clusters extract energy and produce vitamins differently [110].

### 3.4. Functions of the Intestinal Microbiota

The gut microbiota is essential for the metabolism, but also for protective and structural activities.

Metabolic functions include extraction of energy from indigestible dietary polysaccharides and release of vitamins not produced by the human host (vitamin B complex and vitamin K) [111]. In addition, bacteria produce short-chain fatty acids (SCFAs: acetate, propionate and butyrate) and amino acids. In the colonic lumen, the microbiota greatly contributes to the biotransformation of primary to secondary bile acids during their ongoing enterohepatic re-circulation [105]. In turn, in the small intestine, bile acids are important to avoid a significant bacterial colonization [112,113]. Shift towards different patterns of bile acids and bile acid pool may affect cholesterol and glucose homeostasis [114], as well as other metabolic pathways [115,116]. Gases and SCFAs [117] derive from the intraluminal fermentation of undigested carbohydrates by bacteria. SCFAs, in addition, protect the colonic mucosa [117]. Finally, SCFAs influence the lipogenesis in adipose tissue and cholesterol synthesis [118], and have hypocholesterolemic action by inhibiting liver cholesterol synthesis [117,119]. In obese individuals, intestinal microbiota harvests energy more efficiently from carbohydrates otherwise non-digestible. Mechanisms include increased production of SCFA and decreased intestinal expression of *Fiaf* (fasting induced adipose factor), with higher availability of fatty acids to the liver and adipose tissue [99,120,121,122,123].

The intestinal microbiota has a protective role due to a prevention of the gut colonization lumen by pathogens, the contribution in the development of the host immune system (B cell development [124], regulatory T cells, T helper type 1, 2 and 17 cells [125]) and a modulation of inflammatory cytokines. The microbial community prevents infections by pathogens via production of antimicrobial peptides, but also directly competes for metabolic niches [126]. SCFAs per se may modulate the immune system regulating the inflammatory responses [127,128].

The structural activity is mainly secondary to the interactions between the microbiota and the mucus layer which acts as a barrier to inflammatory molecules [129]. The SCFA butyrate improves the colonic defensive border [130]. Further beneficial effects derive from the strong immune-activating properties of microbiota and components such as LPS, peptidoglycans, superantigens, bacterial DNA and heat shock proteins (HSPs).

### 3.5. The Microbiota and the Brain-Gut Axis

The brain-gut axis is a bidirectional pathway involving immune cells and neural pathways [131]. More extensively, the “microbiota-gut-brain axis” is a communication system connecting the intestinal lumen and the brain through immune, endocrine and neuronal pathways. This complex interplay could affect mood, behavior and perception [132,133,134]. The central (CNS) and the autonomous (ANS) nervous system, the neuro-immune and neuro-endocrine systems, the enteric nervous system, and the gut microbiota are all involved in this scenario, where gastrointestinal function can modulate brain signaling [134]. Indeed, gut microbiota can influence anxiety, stress response [135,136,137] and memory function [138]. In animals exposed to infection or affected by inflammation, the gut microbiota modulates the system via a neural protection [138,139,140] and can modulate behavior [141]. Brain development could be regulated via neuronal circuits involved in motor control and anxiety behavior [136]. Indeed, gut bacteria has been reported to interact in a concentration-dependent manner with the brain derived neurotrophic factor (BDNF), a neurotrophin involved in neuronal growth, with modulation of cognitive and emotional behavior [135,136,137]. In addition, a shift in the relative abundance in microbial species or exposure to specific commensal bacteria can interfere with the hypothalamic-pituitary-adrenal (HPA) axis. This pathway influences stress response predisposing to altered mood or behavioral disorders [133,135,142,143]. Microbiota–vagus nerve interaction is also able to affect the interaction between immune, visceral signals and the CNS [140,141,142,144].

## 4. Gene-Bacteria Interplay and Composition of Gut Microbiota in FMF Patients

Autoinflammatory genes, such as *MEFV*, drive an exaggerated innate immune response to various signals in vitro, including microbial products [55]. In parallel, the *NOD2/CARD15* gene is a major susceptibility gene for Crohn’s disease, a chronic, recurrent inflammatory bowel disease (IBD). Similar to the *MEFV* gene, the *NOD2/CARD15* gene is localized to chromosome 16 [145]. Both *MEFV* and *NOD2/CARD15* genes encode similar superfamily proteins, acting as intracellular pattern recognition receptors [146], and likely both regulates cytokine processing, cell apoptosis and inflammation. Patients with Crohn’s disease are carriers of mutated proteins, which sense bacterial products and activate the innate immune response [147]. *NOD2/CARD15* mutations were not associated to an increased susceptibility to develop FMF. However, in a cohort of 103 FMF children, subjects with NOD2/CARD15 mutations had a higher rate of acute scrotum attacks, erysipelas-like erythema, a trend towards higher rates of resistance to colchicine, and a more severe disease, as compared to those without mutations [148].

Xu et al. found that pyrin is a specific immune sensor for bacterial modifications of Rho GTPases, and responds to *Clostridium difficile*, a frequent cause of nosocomial diarrhea. Pyrin does not directly recognize the microbial products but can detect pathogen virulence activity [38]. This finding is relevant for a full comprehension of FMF pathogenesis. On this way, colchicine is the principle therapy for FMF-patients, aimed to prevent acute attacks and complications secondary to chronic inflammation [149]. Colchicine is a fat-soluble alkaloid binding to β-tubulin, hindering its polarization and therefore inhibiting neutrophil chemotaxis and reducing the expression of adhesion molecules. Through these mechanisms, colchicine prevents febrile attacks and controls inflammation in FMF patients. However, 5–10% of FMF patients are non-responders to colchicine [150], possibly due to concomitant diseases (e.g., IBD or vasculitis) [151,152] or to occult infections triggering a reduced drug effectiveness [153,154].

*MEFV* variants are mainly represented by missense mutations in the C-terminal half of the pyrin protein [34,35,36,155]. In homozygous mutant mice expressing a truncated pyrin, the bacterial endotoxin lipopolysaccharide (LPS) induced increased fever and lethality. The mutant pyrin was less effective than the wild-type pyrin in binding to ASC and inhibiting caspase 1 and IL-1β production. Thus, one possibility is that FMF patients become more responsive to transient bacteremia and bacterial pathogens and LPS release, and therefore to systemic inflammatory response [156].

In FMF patients, the presence of a concomitant *Helicobacter pylori* (HP) infection is linked with more severe and more frequent febrile attacks. Of note, HP eradication was associated with beneficial effects (i.e., reduced febrile attacks and cytokine levels) [157,158].

Small intestinal bacterial overgrowth (SIBO), is a condition characterized by the increase of microorganisms in the small bowel exceeding 10^5^ CFU/mL [159,160] and increased bacterial fermentation of a non-adsorbable carbohydrate substrate [161]. The occurrence of SIBO could exacerbate the FMF phenotypic expression. SIBO may generate variable clinical features, ranging from the absolute absence of symptoms to a classical malabsorption syndrome, with dyspepsia, abdominal distension and diarrhea with or without colicky pain, eventually modified by meals and evacuation of stools. The malabsorption and the altered intestinal microbiota might facilitate, in patients with SIBO, the diffusion of products deriving from bacterial metabolism through the blood stream, thus contributing to the dissemination of pathogen associated molecular patterns (PAMPs) [162,163]. This condition might also interfere with a physiological intestinal permeability [164] as well as with the bioavailability of drugs [165]. SIBO could be responsible of unresponsiveness to colchicine, while SIBO decontamination therapy with rifaximin, a non-absorbable antibiotic, contributed to a decrease in FMF attacks [166]. Therefore, SIBO-derived bacterial antigen production or release may promote the secretion of inflammatory cytokines, such as IL-1β, and may sustain a chronic or occult inflammation, leading to an FMF phenotype apparently unresponsive (or hyporesponsive) to colchicine.

Full characterization of gut microbiota in FMF patients is required. Major difficulties derive from phenotypic variations and gene-environment interactions. FMF patients, as compared with healthy subjects, might exhibit a different composition in gut microbiota [167,168]. In principle, the profile of microbial products and metabolites in the human metabolome from FMF patients (in particular the specific profile of long chain fatty acids) might become a marker for the disease [169]. Similarly, increased blood levels of short chain fatty acids appear in the acute phase of the disease, as a consequence of active inflammation [170].

In a series of 19 FMF patients explored during an attack, as compared with healthy controls, a poorer microbiota with loss of diversity has been described, with major shifts in bacterial populations within the Firmicutes, Bacteroidetes and Proteobacteria phyla (i.e., as compared with controls, a lower proportion of Prevotellaceae, Dialister and Prevotella; increase in Porphyromonadaceae, Phascolarctobacterium, aecalibacterium and arabacteroides). In the same subjects, during remission, the amount of Ruminococcus, Megasphaera, Enterobacteriaceae and Acidaminococcaceae was higher than in controls. Conversely, Roseburia was reduced. Thus, host genes may dictate the host-microbiota interaction, with a microbiota profile specific for FMF, and with the most diverse gut bacterial community observed during remission [168]. Additionally, a combined analysis of mutations in the *MEFV* gene and gut bacterial diversity suggested that the described depletion of total numbers of bacteria, loss of diversity and major shifts in bacterial populations depended on the allele carrier status of the host [168].

Different results derive from a more recent study on 41 FMF patients. Data from this series showed specific variations in gut microbiota linked with FMF but, more specifically, a decrease in α-diversity and a significantly altered microbiota composition, with several operational taxonomic units (OUT, i.e., cluster of similar sequence variants of the 16S rDNA marker gene sequence used to distinguish bacteria at the genus level) belonging to the order Clostridiales [167]. Variations with the study of Khachatryan et al. [168] might depend on the different statistical method employed for the analysis (multivariate analysis), a lower number of enrolled subjects in the previous study and the different country of origin of patients [167].

Moreover, Pepoyan et al. [171] observed that M694V/V726A pyrin mutations leading to FMF disease may contribute to gender-specific differences in microbial community structure in FMF patients, although this study analyzed a small number of subjects.

The autoinflammatory state per se can play a critical role in the determination of microbiota variations observed in FMF patients. Armenian FMF patients showed an elevated systemic reactivity against gut microbiota. Inflammatory alterations were also present in the absence of acute attacks, with increased levels of IgG antibodies against commensal microbiota (i.e., *Bacteroides*, *Parabacteroides*, *Escherichia* and *Enteroccocus* antigens) [172]. Another study found a specific association between the presence of AA amyloidosis (i.e., subjects with a complicated disease) and two operational taxonomic units belonging to Clostridiales [167]. This difference does not appear to be attributable to the use of colchicine, the most common drug employed in FMF patients, since this drug, in vitro, does not seem to be able to affect the gut microbiota [173]. Additionally, the oral administration of colchicine in subjects with FMF is not able to normalize the altered profile of microbial long chain fatty acids, microbial products circulating in the systemic metabolome [174]. Conversely, as suggested in other diseases linked with chronic inflammation, it is possible that the appearance of amyloidosis can depend on changes in the gut microbiota [167,175,176,177], independently from genetic factors. Studies exploring gut microbiota in FMF patients report discrepant results [167,168]. Evidence points to a dominant role of environmental factors over host genetics [21]. FMF patients display further variations in the microbiota linked with the presence of AA amyloidosis [167], i.e., when the most severe form of FMF occurs.

Alimov et al. [178] investigated the role of bile acid analogues (BAA) in activating the pyrin inflammasome. Both BAA473, and less potently BAA485, led to IL-18 release from peripheral blood mononuclear cells (PBMCs). Furthermore, BAA473 induced secretion of IL-18 from a human colonic adenocarcinoma cell line and the basolateral side of a human intestinal organoid. Finally, ASC and pyrin were required for IL-1β and IL-18 secretion and colchicine blocked BAA473-mediated inflammasome activation confirming the specific role of pyrin in the process. Increasing evidence is accumulating on the role of the gut microbiota on bile acid bioconversion with interindividual variations driving susceptibility to infections, altered metabolism and immune response [179,180,181,182]. Thus, genetic factors can represent one of several variables determining the gut microbiota profile in subjects with FMF.

## 5. The Role of Environmental Factors and the Determination of Phenotype

Despite the genetic origin, environmental factors might influence the prevalence of different FMF phenotypes. As previously shown, one factor might be the living country [183,184], possibly depending on the combinations of genetics with country-specific factors as dietary habits, deprivation, living conditions and contamination of environmental matrices. These observations are relevant, in particular, in the determination of individual susceptibility to amyloidosis [183].

An analysis on FMF patients from 14 countries demonstrated that the living country, rather than the *MEFV* genotype, was the major factor determining an increased risk of amyloidosis [183]. Furthermore, a comparison between Turkish children with FMF living in Turkey or in Germany showed a more severe course of the disease in those living in Turkey, pointing to the environment as a strong influencer of the FMF phenotype [184]. In this scenario, environmental factors affecting gut microbiota could have a role in determining onset and severity of complications in the context of a monogenic disease of the innate inflammatory pathway. Gut microbiota might also influence the evolution of AA amyloidosis, the most severe complication of FMF [167].

Finally, it is possible that different levels of basal state activation of pyrin, dependent on the *MEFV* genotype, could subtly influence the intestinal homeostasis in the gut conferring an interindividual diverse risk to develop chronic inflammation. In this light, microbiota metabolites are capable of modulating other inflammasomes [185].

## 6. Possible Therapeutic Implications

The link between gut microbiota, FMF acute attacks and FMF complications (i.e., AA amyloidosis), together with evidences pointing to an environmental modulation of gut microbiota, could allow novel therapeutic strategies in FMF patients. One evidence is that a specific probiotic therapy may induce a normalization of serum C-reactive protein (CRP) in FMF patients with high CRP levels during remission [186,187]. This approach appeared to restore the integrity and functionality of the gut microbiota [188].

In particular, *Lactobacillus acidophilus* INMIA 9602 Er-2 strain 317/402, a probiotic strain isolated from feces of a healthy newborn infant [189], produces a small anti-microbial peptide (bacteriocin acidocin LCHV), with a broad spectrum of activity against human pathogens, including methicillin-resistant *Staphylococcus aureus* and *Clostridium difficile* [190]. The strain’s clinically proven positive effects have been confirmed in different studies, also including FMF patients [191]. Interestingly, *Lactobacillus acidophilus* INMIA 9602 Er-2 strain 317/402 was able to reduce not only Enterobacteriaceae, thus bacterial-related intestinal dysbiosis, but also the relative abundance of *Candida albicans*, which is increased in FMF-patients. At the moment there is no evidence on the presence of an altered microbiota composition conditioning the resistance to colchicine in patients with FMF. We recently tested a combination of eight bacterial strains (*Bifidobacterium breve DSM24732^®^, Streptococcus thermophilus DSM24731^®^, Bifidobacterium infantis DSM24737^®^, Bifidobacterium longum DSM24736^®^, Lactobacillus acidophilus DSM24735^®^, Lactobacillus paracasei DSM24733^®^, Lactobacillus plantarum DSM24730^®^, Lactobacillus delbrueckii ssp. bulgaricus DSM24734*) at a concentration of 450 billion bacteria as the De Simone Formulation and available under the trademark Vivomixx^®^ in Europe and Visbiome^®^ in the US. Our preliminary results suggest that this combination, given during the inter-critical period, might improve symptoms in the subgroup of FMF patients carrying *MEFV* variants associated with more severe disease, and partially resistant to colchicine. However, well-designed, large, comprehensive, prospective and definitive studies are missing on the effects of probiotics in FMF patients to prevent the attacks, to reduce symptoms, to ameliorate the efficacy of colchicine and to prevent complications (i.e., amyloidosis).

## 7. Conclusions

The microbiota has an essential role in the host gut and is sensitive to genetic and environmental changes in both health and disease. FMF, as a model of rare inherited monogenic autoinflammatory disease, offers a background of periodic inflammatory changes, with a major involvement of the innate immunity. The microbiota is highly sensitive to such inflammatory changes. In addition, it might govern specific autoinflammatory responses in FMF. FMF symptoms might be sensitive as well, and this emerging topic deserves more attention as a model of environment-genetic interaction. Gut microbiota is likely a key factor in determining the FMF phenotype. In FMF patients, the microbiota abundance and its composition could depend on both genetic and environmental factors with the genome, although it plays a minor role. On the other hand, environmental variables could be critical in shaping the disease severity and complications onset (i.e., AA amyloidosis) in the long term. Further studies need to explore gene–environment interactions in FMF patients. Moreover, possible beneficial effects deriving from external manipulation of gut microbiota require additional investigation on how specific probiotic treatments could improve symptoms and microbiota growth, without reducing the beneficial effects of main therapeutic options in FMF patients.

## Figures and Tables

**Figure 1 genes-11-01041-f001:**
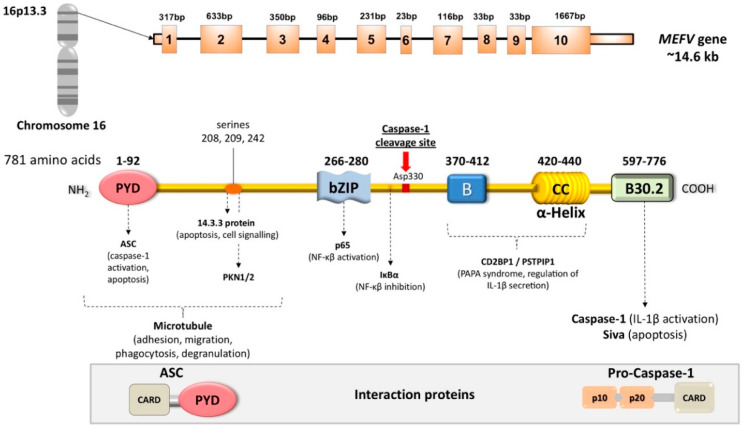
Schematic structure of *MEFV* gene and encoded pyrin (marenostrin) protein. The *MEFV* gene encodes for the pyrin protein (781 amino acids). The most common mutations in Familial Mediterranean Fever (FMF) are in exon 10 encoding the B30.2 domain. The most important interaction partners appear below the pyrin structure. ASC and Pro-Caspase 1 are also drawn. Pyrin structure includes five different domains, each one responsible for protein-protein interaction, and each domain has a role in the regulation of innate response. From left to right, PYRIN (PYD) domain (residues 1–92) interacts with ASC (apoptosis-associated speck-like protein containing a CARD—caspase-recruitment domain). bZIP transcription factor basic domain (residues 266–280) interacts with the p65 subunit (transcription factor p65) of NF-κB, and IκB-α. The B-box zinc finger domain (residues 375–412) and α-helical (CC, coiled-coil) domain (residues 420–440) likely influence the oligomerization of pyrin, and interact with the PAPA protein (also named PSTPIP1, proline serine threonine phosphatase-interacting protein, also known as CD2BP1 involved in the organization of the cytoskeleton) and the regulation of IL-1β secretion. The B30.2 domain (PRYSPRY) (residues 597–776) is the most important, and interacts with caspase-1 and the proapoptotic protein Siva. Further pyrin interactions include binding to microtubules (starting from the *N*-terminal to bZIP), interaction with 14.3.3 (14-3-3 protein), and with the PKN1/2 (serine-threonine kinases PKN1 and PKN2) at the three serine residues 208, 209, 242 between PYD and bZIP. The position of Asp330 between bZIP and the B-box indicates the caspase-1 cleavage site. Mutations in the B30.2 domain tend to be transmitted in an autosomal-recessive fashion. Mutations in exons 2, 3 and 5 generally exhibit autosomal-dominant pattern of inheritance [24,34,35,36,41,43,48,49,50,51,52,53,54,55,56,57,58].

**Figure 2 genes-11-01041-f002:**
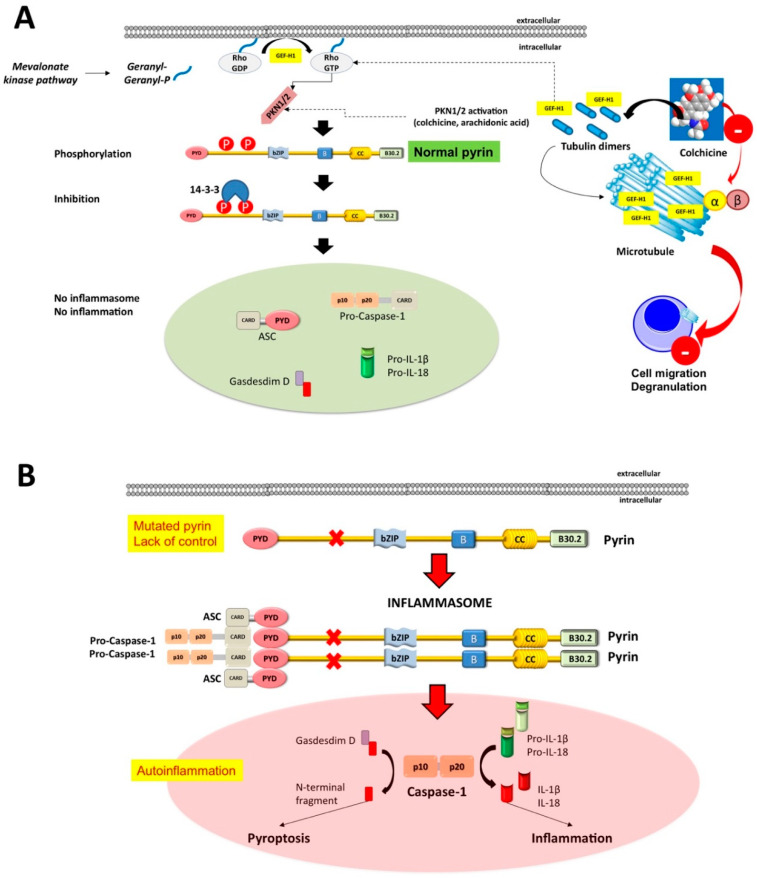
Mechanisms underlying the assembly of the pyrin inflammasome. (**A**) With a normally functioning pyrin (i.e., when pyrin is non-mutated), the mevalonate kinase pathway provides geranyl-geranyl phosphate and, together with release of GEF-H1 (increased by colchicine acting as the inhibitor of the microtubule polymerization), activates RhoA. PKN1 and PKN2 are effector kinases of RhoA mediating the phosphorylation of pyrin and binding to the inhibitory proteins 14-3-3. Inhibition of pyrin can increase with agents activating PKN1/2 or following the release of GEF-H1 (i.e., colchicine). (**B**) If the pyrin phosphorylation decreases, (i.e., in FMF patients lacking the control by pyrin/marenostrin caused by pathogenic *MEFV* variants), with low GEF-H1 or defective function of the MVK-pathway, the activation of PKN1/2 also decreases. This step results in pyrin inflammasome activation and release of mature IL-1β and IL-18. The plasma membrane pore-forming *N*-terminal fragment of gasdermin D facilitates IL-1β and IL-18 release. Appropriate stimuli can also lead to the assembly of the inflammasome. The first step is the PYD-PYD homotypic interaction of ASC resulting in oligomerization into ASC specks. Pro-caspase-1 is recruited because of CARD-CARD interaction with ASC. This step anticipates the auto-cleavage of pro-caspase-1 into active caspase-1 tetramers (p10/p20) governing the transformation of pro-IL-1/18 into mature IL-1/18. The pyroptosis mediated by Gasdermin D also contributes to cytoplasmic enrichment with IL-1/18, and further reinforces the inflammatory pathway. Colchicine inhibits the polymerization of intracellular β-tubulin by forming colchicine-tubulin complexes via contact of A and C rings with the C domain of the tubulin β-subunit. These supramolecular interactions block the dockage of tubulin into the (+) ends of microtubules (cytoskeleton), thus preventing inflammasome activation in neutrophils and monocytes. The colchicine-dependent inhibition of tubulin also efficiently affects the migration and degranulation of white blood cells [43,50].

**Figure 3 genes-11-01041-f003:**
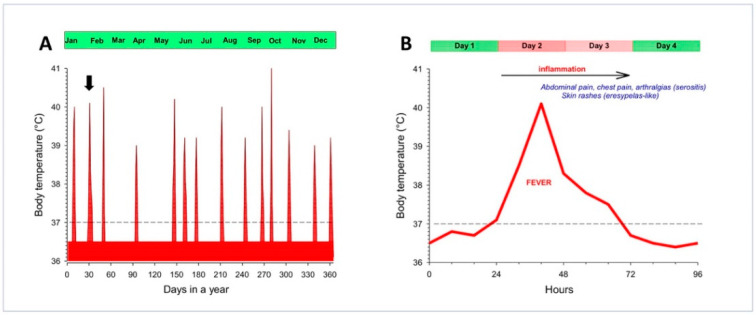
Schematic appearance of time-dependent changes of body temperature in Familial Mediterranean Fever (FMF) before starting the treatment with colchicine. The profile refers to a typical case belonging to a cluster of families identified in the region of Apulia, Italy [24]. (**A**) Frequency of febrile attacks in a year. The dotted horizontal line is placed at 37 °C (cut-off value). In between attacks, the temperature has been conventionally set at 36.5 °C. This patient reported a total of 14 attacks in a year. The black arrow indicates the febrile attack described in panel B. (**B**) The single febrile attack lasts about 48 h and is associated with a major auto-inflammatory status and symptoms. Adapted from Portincasa et al. Familial Mediterranean fever: a fascinating model of inherited auto inflammatory disorder. Eur J Clin Invest 43, 1314-1327 (2013) with permission from John Wiley & Sons Ltd. [24].

**Figure 4 genes-11-01041-f004:**
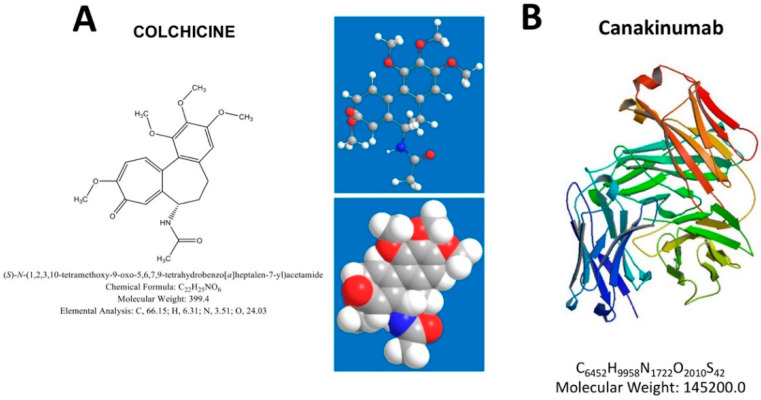
Therapeutic agents effective in Familial Mediterranean Fever (FMF). (**A**) Colchicine: chemical structure, IUPAC names, chemical formula, molecular weight and three-dimensional (3D) structures. (**B**) Canakinumab: 3D structure, chemical formula and molecular weight.

**Figure 5 genes-11-01041-f005:**
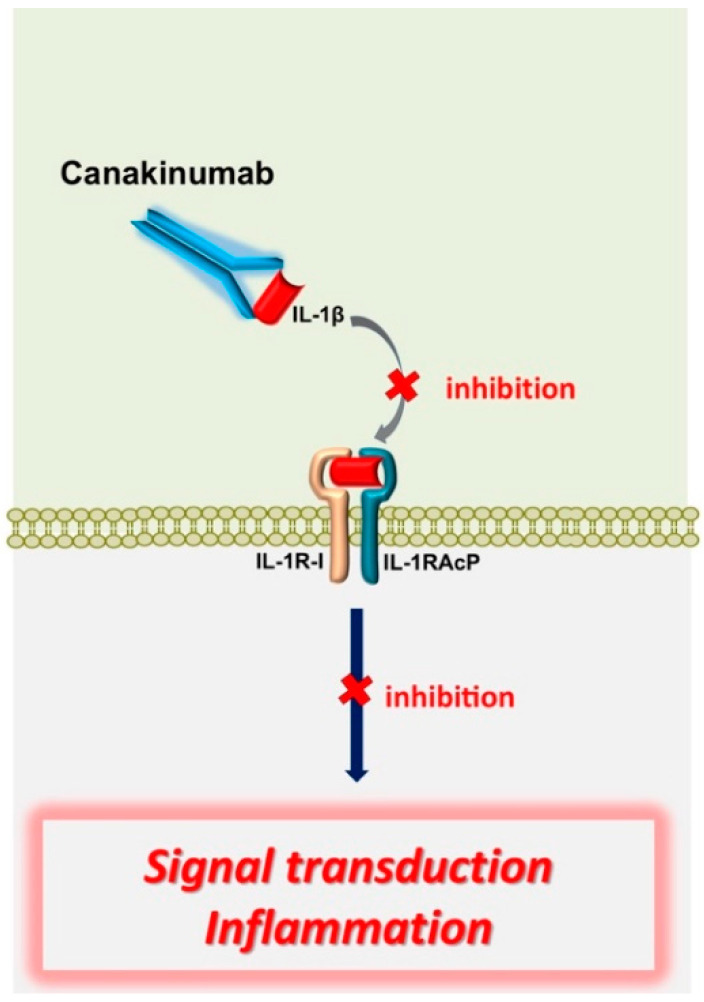
Mechanism of action of biologic agent canakinumab in FMF. Canakinumab is a fully human selective anti-IL-1β monoclonal antibody and binds human IL-1β. Subsequent binding of IL-1β to the IL-1R is inhibited with prevention of intracellular signal transduction and further proinflammatory events. Abbreviations: IL, interleukin; IL-1R-I, interleukin-1 receptor, type1; IL-1RAcP, IL-1 receptor accessory protein. Adapted from [49,50,51].

**Figure 6 genes-11-01041-f006:**
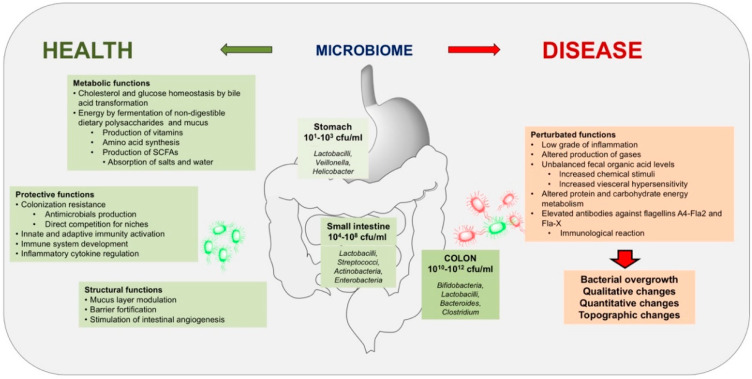
Distribution and function of microbiota in human intestine during health or disease. The bacterial community is the core component of microbiota in the gastrointestinal tract and there are eukaryotic microbes. In healthy subjects, microbial density and types vary across the gastrointestinal tract, with density increasing dramatically in oro-aboral direction. Microbial composition also changes. The crosstalk between bacteria and host contribute to maintain physiological metabolic, protective, and structural functions. In disease, shift of relative abundance in microbial species or quantitative variations of microbiota occur, generating chronic low-grade inflammation, atypical gases and fecal levels of organic acids, impaired metabolism of proteins and carbohydrates and immunological responses [2,102,103,104,105,106].

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
