# Peer review of "Gut Microbiota between Environment and Genetic Background in Familial Mediterranean Fever (FMF)"

_genes, 2020, doi:10.3390/genes11091041_

Round 1

Reviewer 1 Report

This manuscript provides a review on the pathology of Familial Mediterranean Fever (FMF) and the role of interaction between host and microbiota on the disease. It is related to multiple disciplines including cytology, pathology, and microbiology. This comment mainly focused on the microbiology part.

Line 45. Studies focusing on human twins [7], murine quantitative trait loci [8], and genome-wide associations [9], suggest that genes can drive the composition of gut microbiome.

Please specify here what genes can drive the composition of gut microbiome. Is it the host genomic profile or the genes of gut bacterial species?

Line 64. The estimated number of patients with FMF is 150,000 …

Is this number the amount of FMF cases over the world? Please specify it in the text.

Line 85. which express a variety of pattern recognition receptors (PRRs).

Please provide the abbreviation at the first appearance four lines ahead rather than here.

Line 221. … is associated with further microbial changes and diversity and already in the infant the gut microbiota resembles that of the adult.

Please check the grammar of this sentence.

Line 229. … studies of metagenomes to compare assembled sequences to reference databases.

Consider rephrasing it to ‘metagenomic assembly, annotation and comparison’. Also, you may want to add the technique genome-resolved metagenomic analysis which provides essential genetic information of single species.

Line 236. … a large global number of species-level phylotypes (> 1,000) but phylogenetic individual diversity is low …

What are the phylotypes (>1000)? Amplicon sequence OTUs? Please further explain the reason for it in the text.

Line 249. The microbiome is the core of bacterial community in the gastrointestinal tract.

Bacterial community is the core component of microbiome in the gastrointestinal tract and there are Eukaryotic microbes.

Line 253. In disease, qualitative and/or quantitative alterations of microbiome occur …

Do you mean the presence and absence of bacterial species by ‘qualitative alteration’? When a bacterial species is ‘absent’, it is probably in the rare biosphere with extremely low abundance that is not even detectable by current technologies, but it is there. So, I think ‘shift of relative abundance in microbial species’ may be more proper here.

Line 293. Structural activity.

Is it a subtitle or a sentence here?

Author Response

Answers to reviewer 1

This manuscript provides a review on the pathology of Familial Mediterranean Fever (FMF) and the role of interaction between host and microbiota on the disease. It is related to multiple disciplines including cytology, pathology, and microbiology. This comment mainly focused on the microbiology part.

- Line 45. Studies focusing on human twins [7], murine quantitative trait loci [8], and genome-wide associations [9], suggest that genes can drive the composition of gut microbiome. Please specify here what genes can drive the composition of gut microbiome. Is it the host genomic profile or the genes of gut bacterial species?

Many thanks for this observation. The sentence has been changed as follows:

“Studies focusing on human twins [15], murine quantitative trait loci [16], and genome-wide associations [17], suggest that the host genomic profile can shape the composition of gut microbiota..”

- Line 64. The estimated number of patients with FMF is 150,000 … Is this number the amount of FMF cases over the world? Please specify it in the text.

The sentence has been changed as follows:

The estimated number of subjects with FMF in the world is 150,000, making FMF the most common autoinflammatory disease worldwide [21].”

- Line 85. which express a variety of pattern recognition receptors (PRRs).

Please provide the abbreviation at the first appearance four lines ahead rather than here.

Thanks for this observation. The text has been changed according to this suggestion.

- Line 221. … is associated with further microbial changes and diversity and already in the infant the gut microbiota resembles that of the adult.

Please check the grammar of this sentence.

The sentence has been changed as follows:

“Feeding with solid food in the infant is followed by changes in the composition of gut microbiota, with features resembling those of the adult [83]..”

- Line 229. … studies of metagenomes to compare assembled sequences to reference databases.

Consider rephrasing it to ‘metagenomic assembly, annotation and comparison’. Also, you may want to add the technique genome-resolved metagenomic analysis which provides essential genetic information of single species.

Thank you for these interesting suggestions. The text has been changed as follows:

“Techniques include sequencing and phylogenetic micro-arrays [93,94], metagenomic assembly, annotation and comparison [3], genome-resolved metagenomics..”

- Line 254. … a large global number of species-level phylotypes (> 1,000) but phylogenetic individual diversity is low …

What are the phylotypes (>1000)? Amplicon sequence OTUs? Please further explain the reason for it in the text.

The sentence has been removed, since it was redundant.

- Line 249. The microbiome is the core of bacterial community in the gastrointestinal tract.

Bacterial community is the core component of microbiome in the gastrointestinal tract and there are Eukaryotic microbes.

Many thanks for this observation. The text has been changed as suggested

- Line 253. In disease, qualitative and/or quantitative alterations of microbiome occur …

Do you mean the presence and absence of bacterial species by ‘qualitative alteration’? When a bacterial species is ‘absent’, it is probably in the rare biosphere with extremely low abundance that is not even detectable by current technologies, but it is there. So, I think ‘shift of relative abundance in microbial species’ may be more proper here.

We thank the reviewer for this observation. The definition of the reviewer is appropriate. The sentence has been changed as follows:

“In disease, shift of relative abundance in microbial species or quantitative variations of microbiota occur, generating chronic low-grade inflammation, atypical gases and faecal levels of organic acids, impaired metabolism of proteins and carbohydrates, and immunological responses [2,101-105].”

- Line 293. Structural activity. Is it a subtitle or a sentence here?

The sentence has been rephrased as follows:

“The structural activity is mainly secondary to the interactions between the microbiota and the mucus layer which acts as a barrier to inflammatory molecules [128].”

Reviewer 2 Report

The authors in the article entitled “Gut microbiota between environmental and genetic background in Familial Mediterranean Fever (FMF)” have review the literature and complied valuable information regarding the interplay between host gut microbiota and this auto-inflammatory disease. The authors could nicely reference recent and relevant studies to support their statements. I do have few comment that are indicated in the following.

L27. It is the abstract, you have written “… qualitative and quantitative changes of bacterial population”. Throughout the whole text, I could not find a study claiming so, please correct this appropriately (intermix populations with communities) or indicate relevant study.

L40. Please provide a reference for each of these mentioned aspect.

L52. Please homogenize throughout the text microbiome or microbiota. If microbiome has a different meaning here, please indicate.

L104. It is not very clear what the authors would like to convoy by “wild-type”. Could you please further explain the meaning here.

L142. What does mean a normal condition?

L215. Please cite the appropriate paper inhere, ref 66 is also referencing other papers.

L421. I am a bit puzzled by the fact that living country that influence FMF phenotypes. Does this mean by country the interacting dietary habits and genetics? It would be important to further explain this factor.

L439. What looks like a normal CRP? Please specify here.

L455. Are the patients with resistant to colchicine phenotype having a different microbiota that could reduce the efficacy of the molecule? Would be a therapeutic way of altering the microbiota to ameliorate the efficacy of the drug?

Author Response

Answers to reviewer 2

The authors in the article entitled “Gut microbiota between environmental and genetic background in Familial Mediterranean Fever (FMF)” have review the literature and complied valuable information regarding the interplay between host gut microbiota and this auto-inflammatory disease. The authors could nicely reference recent and relevant studies to support their statements. I do have few comments that are indicated in the following. 

- L27. It is the abstract, you have written “… qualitative and quantitative changes of bacterial population”. Throughout the whole text, I could not find a study claiming so, please correct this appropriately (intermix populations with communities) or indicate relevant study.

The sentence has been changed as follows:

“Growing evidences show a possible link between the microbiota and FMF, through a shift of the relative abundance in microbial species.”

- L40. Please provide a reference for each of these mentioned aspect.

We thank the reviewer for this suggestion. The sentence has been changed as follows:

“Examples include inflammatory bowel disease [7], irritable bowel syndrome  [8, periodontal disease {Curtis, 2020 #22277], atherosclerosis [9], rheumatoid arthritis [10], diabetes mellitus [11], obesity [12], allergy [13], and colonic cancer [14].” .

- L52. Please homogenize throughout the text microbiome or microbiota. If microbiome has a different meaning here, please indicate. 

The term “microbiome” has been replaced with “microbiota” throughout the manuscript.

- L104. It is not very clear what the authors would like to convoy by “wild-type”. Could you please further explain the meaning here.

The sentence has been changed as follows:

“In a normal situation, pyrin senses the inactivation of …”

- L142. What does mean a normal condition? 

The sentence has been changed as follows:

“A) With a normally functioning pyrin (i.e. when pyrin is non-mutated),..”

- L215. Please cite the appropriate paper inhere, ref 66 is also referencing other papers. 

Many thanks to the reviewer for this observation. The correct paper has been now cited and the sentence has been changed as follows:

“The in utero environment is not sterile, with a maternal-fetal transmission of microbiota occurring early, during pregnancy [74][75].”

- L421. I am a bit puzzled by the fact that living country that influence FMF phenotypes. Does this mean by country the interacting dietary habits and genetics? It would be important to further explain this factor. 

Evidences reporting different FMF phenotypes in different countries point to possible gene-environment interactions. The paragraph has been now re-written as follows:

” As previously shown, one factor might be the living country [182,183], possibly depending on the combinations of genetics with country-specific factors as dietary habits, deprivation, living conditions, contamination of environmental matrices.”

- L439. What looks like a normal CRP? Please specify here.

The sentence has been reformulated as follows:

” One evidence is that a specific probiotic therapy may induce a normalization of serum C-reactive protein (CRP) in FMF patients with high CRP levels during remission [185,186].”

- L455. Are the patients with resistant to colchicine phenotype having a different microbiota that could reduce the efficacy of the molecule? Would be a therapeutic way of altering the microbiota to ameliorate the efficacy of the drug?

At the moment no evidence is available on possible relationships between a specific microbiota profile and the therapeutic efficacy of colchicine. The paragraph has been changed as follows:

“At the moment there is no evidence on the presence of an altered microbiota composition conditioning the resistance to colchicine in patients with FMF. We recently tested a combination of eight bacterial strains (Bifidobacterium breve DSM24732®, Streptococcus thermophilus DSM24731®, Bifidobacterium infantis DSM24737®, Bifidobacterium longum DSM24736®, Lactobacillus acidophilus DSM24735®, Lactobacillus paracasei DSM24733®, Lactobacillus plantarum DSM24730®, Lactobacillus delbrueckii ssp. bulgaricus DSM24734) at a concentration of 450 billion bacteria as the De Simone Formulation and available under the trademark Vivomixx® in Europe and Visbiome® in the US. Our preliminary results suggest that this combination given during the intercritical period, might improve symptoms in the subgroup of FMF patients carrying MEFV variants associated with more severe disease, and partially resistant to colchicine. However, well-designed, large, comprehensive, prospective and definitive studies are missing on the effects of probiotics in FMF patients to prevent the attacks, to reduce symptoms, to ameliorate the efficacy of colchicine and to prevent complications (i.e. amyloidosis).”